# Inhibition of Abl Kinase by Imatinib Can Rescue the Compromised Barrier Function of 22q11.2DS Patient-iPSC-Derived Blood–Brain Barriers

**DOI:** 10.3390/cells12030422

**Published:** 2023-01-27

**Authors:** Yunfei Li, Zhixiong Sun, Huixiang Zhu, Yan Sun, David B. Shteyman, Sander Markx, Kam W. Leong, Bin Xu, Bingmei M. Fu

**Affiliations:** 1Department of Biomedical Engineering, The City College of the City University of New York, New York, NY 10031, USA; 2Department of Psychiatry, Columbia University, New York, NY 10032, USA; 3Department of Biomedical Engineering, Columbia University, New York, NY 10027, USA

**Keywords:** blood–brain barrier, imatinib, DiGeorge Syndrome, endothelial glycocalyx, tight junction, permeability, Abl/CRKL signaling pathway, transcription network

## Abstract

We have previously established that the integrity of the induced blood–brain barrier (iBBB) formed by brain microvascular endothelial cells derived from the iPSC of 22q11.2 DS (22q11.2 Deletion Syndrome, also called DiGeorge Syndrome) patients is compromised. We tested the possibility that the haploinsufficiency of CRKL, a gene within the 22q11.2 DS deletion region, contributes to the deficit. The CRKL is a major substrate of the Abl tyrosine kinase, and the Abl/CRKL signaling pathway is critical for endothelial barrier functions. Imatinib, an FDA-approved drug, inhibits Abl kinase and has been used to treat various disorders involving vascular leakages. To test if imatinib can restore the compromised iBBB, we treated the patient’s iBBB with imatinib. After treatment, both trans-endothelial electrical resistance and solute permeability returned to comparable levels of the control iBBB. Correspondingly, changes in tight junctions and endothelial glycocalyx of the iBBB were also restored. Western blotting showed that imatinib increased the level of active forms of the CRKL protein. A transcriptome study revealed that imatinib up-regulated genes in the signaling pathways responsible for the protein modification process and down-regulated those for cell cycling. The KEGG pathway analysis further suggested that imatinib improved the gene expression of the CRKL signaling pathway and tight junctions, which agrees with our expectations and the observations at protein levels. Our results indicate that the 22q11.2DS iBBB is at least partially caused by the haploinsufficiency of CRKL, which can be rescued by imatinib via its effects on the Abl/CRKL signaling pathway. Our findings uncover a novel disease mechanism associated with 22q11.2DS.

## 1. Introduction

Schizophrenia is a severe neuropsychiatry disorder. Patients with schizophrenia require long-term medical care. The average cost for patient care is estimated to be USD 727,178 per patient in 2022 [1]. Therefore, efforts to elucidate the mechanisms of the disease are an urgent need for developing an effective treatment for this complex genetic disorder. However, due to the heterogeneous nature of schizophrenia, the elucidation of its etiology and pathophysiology is difficult [2]. 22q11.2 deletion syndrome (22q11.2DS) is a genetic disorder caused by hemizygous microdeletion of chromosome 22 [3], which leads to variable developmental delays that affect the central nervous system and cardiovascular system [4]. Importantly, about one-third of 22q11.2 deletion carriers develop schizophrenia or schizoaffective disorder, which is roughly 30 times higher than the general population. Therefore, 22q11.2 deletion has been considered to be a strong genetic risk factor for schizophrenia [4]. Elucidating its contribution to neuronal and vascular dysfunctions is thus important to obtain critical insights into the disease mechanisms of schizophrenia.

Our prior study showed that the 22q11.2DS iBBB (induced blood–brain barrier) generated from patient iPSC-derived human brain microvascular endothelial cells (HBMECs) is compromised in its structure and function [5]. Blood–brain barrier (BBB) dysregulation is one of the widely observed phenotypes in schizophrenia and other neuropsychiatric disorders [6,7]. For example, abnormalities in neurovascular unit (NVU) components have been reported in the frontal cortex of the postmortem brains of schizophrenia and autism patients [8,9,10,11]. Of particular note, endothelial tight junction signaling has been shown to be altered in patients with schizophrenia, in which 12 out of 21 tight-junction-related genes are reduced [12]. During our further investigation, we confirmed that the Crk-like (*CRKL*) gene, which encodes an adapter protein belonging to the Crk family, is in the deletion region of the 22q11.2 DS [13] and is reduced in the 22q11.2DS iBBB at both mRNA and protein levels [5].

CRKL is a 39kDa adapter protein [14,15]. It has been shown that active CRKL is recruited to p130Cas, which has multiple protein–protein interaction domains responsible for forming the basic skeleton of tight junctions [16]. Moreover, CRKL interacts with a set of small GTPases, which are critical in regulating cell adhesion and junction stability [17]. Interestingly, CRKL is a key downstream effector of the Abl kinase, which participates in regulating VEGF-mediated endothelial barrier function [17]. The Abl/CRKL signaling pathway is also important in regulating cellular responses such as cytoskeletal remodeling, adhesion, and migration [18]. CRKL is predominantly phosphorylated and inactivated by Abl tyrosine kinases [19]. The abnormal Abl tyrosine kinase activity that affects the phosphorylation of CRKL has been previously observed in leukemogenesis [20]. These prior studies allow us to speculate that the inhibition of Abl kinase activity might reduce CRKL phosphorylation to restore sufficient active/unphosphorylated form of the CRKL protein to rescue the compromised BBB in 22q11.2 DS patients.

Imatinib (STI-571) is a small molecule pharmacological inhibitor blocking the ATPase activity of the Abl kinases [21]. Since being approved by the FDA (Food and Drug Administration) in 2001, imatinib (brand names Gleevec and Glivec by Novartis) has been widely applied for cancer treatment including acute lymphoblastic leukemia, Bcr-Abl positive chronic myeloid leukemia, and gastrointestinal stromal tumors [22,23]. Imatinib has recently been found to attenuate vascular leakage induced by inflammatory agents, vascular endothelial growth factor (VEGF), lipopolysaccharide and oxidative stress [17,24,25,26]. It can also protect against edema, including brain edema after stroke and pulmonary edema [27,28], restore BBB integrity and decrease intracerebral hemorrhage in murine models [29]. We thus chose imatinib as our inhibitor for Abl kinases to test our hypothesis. 

To test if the inhibition of Abl by imatinib can rescue compromised BBB function and structure in previously observed 22q11.2DS patients [5], after iBBBs were generated from HBMECs derived from the hiPSCs of three patients with 22q11.2DS (DEL) and three age- and sex-matched healthy controls (WT), we treated these iBBBs with imatinib and compared TEER, permeability to a large molecule, dextran-70k, as well as tight junction proteins, ZO-1 and claudin-5, and endothelial glycocalyx (heparan sulfate) with and without treatments. We also investigated the molecular mechanisms by which imatinib modulates BBB permeability using Western blotting and RNA sequencing. 

## 2. Materials and Methods

### 2.1. Generation of Induced BBB (iBBB) and Measurement of Its Trans-Endothelial Electrical Resistance (TEER)

As in our previous study [5], two pairs of 22q11.2DS case (DEL1 and DEL2)/control (WT1 and WT2) hiPSC lines were obtained from the NIMH Repository and Genomics Resource (http://www.nimhstemcells.org/, accessed on 26 April 2018) [30]. One pair of hiPSC lines (DEL3 and WT3) was generated at the Columbia Stem Cell Core via non-integrating Sendai virus-based reprogramming [31] of monocytes from a donor with 22q11.12DS and a healthy sibling control. After confirming the stemness of the hiPSC lines, we differentiated them into HBMECs following the protocols in [32,33]. 

To generate an iBBB, human-plasma-derived fibronectin (100 μg/mL) and human-placenta-derived collagen IV (400 μg/mL) (Sigma-Aldrich, St. Louis, MO, USA) were first added in the Transwell filter (Falcon, Corning, NY, USA) with a 0.4 μm pore transparent PET membrane (0.9 cm^2^ bottom area) for coating at 37 °C overnight in an incubator [32,33]. Then, HBMECs were detached from the culture dish and seeded on the Transwell insert at a density of ~50 k/cm^2^ [5]. Human Endothelial SFM (Gibco, Thermo Fisher Scientific, Waltham, MA, USA) supplemented with 1% platelet-poor plasma-derived serum (PDS, Alfa-Aesar, Ward Hill, MA, USA), 2% B-27 (Gibco, Thermo Fisher Scientific, Waltham, MA, USA), 10 μM RA (retinoic acid) (Sigma-Aldrich), 50U/mL penicillin–streptomycin (Gibco, Thermo Fisher Scientific, Waltham, MA, USA), 20 ng/mL bFGF (Sigma-Aldrich) was used for culturing HBMECs [5]. Then, 10 μM of Y-27623 dihydrochloride (Tocris Bioscience, Bristol, UK) was added one day after seeding to promote the cell survival and removed on the following day. The HBMEC monolayer reached confluency in 4–5 days and formed an iBBB in 6–7 days. Before confluency, cells were observed once in two days by a bright-field microscope. After confluency, the TEER of the monolayer was measured every day until no change was observed in at least two consecutive days, in which case the iBBB was considered generated. A chopstick-shaped Volt/Ohm Meter (EVOM2™, World Precision Instruments, Sarasota, FL, USA) was used to measure the TEER of the iBBB. To obtain the actual TEER of the iBBB, the background TEER, which is the TEER of the blank Transwell filter, was subtracted from the total TEER (iBBB + blank filter). The seeding density of 50 k/cm^2^ and culturing time were optimal for our purpose (see Section 4 and Appendix A).

### 2.2. Determination of Solute Permeability (P) of iBBB

To measure the solute permeability (P) of the iBBB, the upper compartment of the Transwell filter was loaded with 0.5 mL of 10 μM FITC–Dex 70k (Dextran-70k, MW 70 kD, Sigma-Aldrich) in 10 mg/mL bovine serum albumin (BSA, Sigma-Aldrich) in a Ringer solution [34]. Then, 1.5 mL of the same solution without FITC-Dex 70k was added to the lower compartment of the Transwell filter. The samples of 50 μL were taken every 10 min for 90 min from the lower compartment and replaced with the same amount of the BSA-Ringer solution. The intensity of the sample solution with FITC-Dex 70k was measured by using a SpectraMax M5 microplate reader (Molecule Devices, San Jose, CA, USA). The permeability $\left(P_{m}\right)$ to Dex-70k was calculated by using Equation (1) [34,35,36],
(1)$$P_{m}=\frac{\frac{\Delta I_{L}}{\Delta t}\times V}{I_{U}\times A}$$
where $\frac{\Delta I_{L}}{\Delta t}$ is the increased rate of the intensity of the fluorescent solution in the lower compartment during the time interval Δt, $I_{U}$ is the fluorescence intensity in the upper compartment, *V* is the solution volume in the lower compartment, and A is the area of the filter membrane. Calibration experiments for the concentration vs. intensity for the FITC-Dex 70k were performed to ensure that the concentration was linearly related to the intensity of the solution used in our study. $P_{iBBB}$ to Dex-70k was calculated by using Equation (2),
(2)$$\frac{1}{P_{m}}=\frac{1}{P_{b}}+\frac{1}{P_{iBBB}}$$
where $P_{m}$ is the measured permeability of both the iBBB and the Transwell filter, $P_{b}$ is that of the blank transwell filter and $P_{iBBB}$ is the permeability of the iBBB.

### 2.3. Immunostaining of Tight Junction Proteins and Glycocalyx of iBBB

*Immunostaining of tight junction proteins (ZO-1 and claudin-5)* [5,32,37]: The generated iBBB was first rinsed with DPBS (Dulbecco’s Phosphate Buffered Saline, Corning, Corning, NY, USA) 3 times, then fixed with 2% paraformaldehyde (Polyscience, Warrington, PA, USA) and 0.1% glutaraldehyde (Sigma-Aldrich) for 20 min and blocked with 10% normal goat serum (NGS, Jackson Immuno Research, West Grove, PA, USA) in 0.1% Triton X-100 (Sigma-Aldrich) for 1 h at room temperature (RT). For ZO-1 and claudin-5 labeling, the iBBB was first incubated with ZO-1 polyclonal antibody (1:100, 40–2200, rabbit; Invitrogen, Thermo Fisher Scientific, Waltham, MA, USA) or with claudin-5 monoclonal antibody (4C3C2) (1:200, 35–2500, mouse; Invitrogen) at 4 °C overnight. After washing 3 times with DPBS, goat anti-rabbit antibody (Alexa Fluor™ 488, 1:200, A-11034; Invitrogen) or goat anti-mouse IgG (Alexa Fluor™ 488, 1:200, A-11001; Invitrogen) was incubated with the iBBB for 1 h at RT. After washing 3 times with DPBS, the iBBB was mounted with Fluoromount g with DAPI (SouthernBiotech™, Birmingham, AL, USA) and made into slides for confocal imaging.

*Immunostaining glycocalyx* [38,39]: We labeled heparan sulfate (HS), the most abundant glycosaminoglycan of the endothelial glycocalyx (EG), to quantify the EG [40,41]. After washing 3 times with 10 mg/mL BSA in PBS, the iBBB was fixed with 2% paraformaldehyde (Polyscience) and 0.1% glutaraldehyde (Sigma-Aldrich) for 20 min at RT. The iBBB was then blocked with 2% NGS in PBS for 30 min at RT, followed by labeling HS with anti-heparan sulfate antibody (1:100, 10E4 epitope, mouse, Amsbio, Abingdon, UK) at 4 °C overnight. After rinsing with DPBS 3 times, iBBB was incubated with goat anti-mouse IgG (Alexa Fluor™ 488, 1:200, A-11001; Invitrogen) for 1 h at RT. Finally, the iBBB was mounted by Fluoromount g (SouthernBiotech™) with DAPI after DPBS washing and made into slides for confocal imaging. 

### 2.4. Quantification of ZO-1, Claudin-5 (CLN-5) and HS (Glycocalyx) 

The iBBB samples were scanned by using a Zeiss LSM 800 confocal laser scanning microscope with a 40×/NA1.3 oil immersion objective lens. Three fields from each sample were taken randomly for scanning. For the imaging of ZO-1 and CLN-5, the scanning field was 160 μm × 160 μm (2048 × 2048) with a z-stack of 50–60 images with a 0.2 µm step size. For glycocalyx, the scanning field was 320 μm × 320 μm (2048 × 2048) with a z-stack of 20–30 images with a 0.49 μm step size. The orthogonal projection of the images and the intensity quantification for ZO-1, CLN-5 and HS were performed using Zeiss ZEN 3.0 and NIH ImageJ [5,34]. We used our previous approach [34,42] to quantify the tight junction proteins of ZO-1 and CLN-5. Briefly, the averaged intensity profile from the 3–5 perpendicular lines (~3 µm) equally distributed along each junction between adjacent ECs was determined for that junction. Then, 20–30 junctions were randomly selected for each experiment, and 60–90 junctions from 3 independent experiments were determined for each case. Because we performed all the experiments on the control (WT) and patients (DEL) for each pair simultaneously, we used the averaged peak intensity of the junction protein from the control WT samples for the normalization in each WT–DEL pair. To quantify the glycocalyx, 3 fields (each field 320 μm × 320 μm) were measured for each sample. Three samples, each from 3 independent experiments (separate differentiations) were analyzed for each case. The average intensity of the control WT samples was used to normalize the HS intensity in the same pair of WT/DEL.

### 2.5. Western Blotting 

The total proteins were extracted using tissue protein extraction buffer (Thermo Scientific, Ref: 78510) containing proteinase inhibitors (Complete Mini, Roche, Ref: 11836170001, Roche, Basel, Switzerland) according to the manufacturer’s instructions. Briefly, HBMECs were washed twice with DPBS solution and then incubated in the lysis buffer for 10 min on ice and vortexed vigorously. The samples were spun down at 10,000 g and the supernatant was stored at −80 °C for downstream analysis. The extracted protein was separated on a NuPAGE 4 to a 12% gradient gel (Invitrogen, Ref: NP0322). Primary antibodies (Phospho-CRKL (Tyr207) (E9A1U) Cell Signaling #34940, 1:800; CRKL (D4G7G) Cell Signaling #38710, 1:800, Cell Signaling Technology, Danvers, MA, USA) were bound by HRP- conjugated secondary antibodies and visualized using the Azure c600 imaging system (Azure Biosystems, Dublin, CA, USA). The images were quantified using Image Studio Lite Software (Licor Inc., Lincoln, NE, USA), and actin protein (b-Actin AbCam # ab227387, 1:6000, Abcam Inc., Cambridge, UK) was used as an internal input control to normalize all measurements.

### 2.6. Total RNA Isolation and Bulk RNA Sequencing

Twelve iPSC-derived HBMEC samples (3 WT without imatinib treatment, 3 WT with imatinib treatment, 3 DEL without imatinib treatment, and 3 DEL with imatinib treatment) were collected from the Transwell iBBB model after the functional test (TEER) was completed for each sample [5]. Briefly, the cell culture medium was removed, and the cell monolayer in the Transwell filter was washed twice with DPBS at RT. Then, 500 μL QIAzol lysis reagent (Qiagen, Germantown, MD, USA) was added, and the cells were thoroughly lysed by pipetting up and down the lysis reagent until there was no cell debris. The lysed cells were transferred to a microcentrifuge tube for each sample. The total RNA was extracted using the miRNeasy kit (Qiagen) according to the instructions of the manufacturer. The concentration and purity of each sample were determined using a spectrophotometer (ND1000; Nanodrop, Thermo Scientific), and the quality of the RNA was confirmed (with RNA integrity index (RIN) score > 9) by using Microchip Gel Electrophoresis (Agilent Technologies, Santa Clara, CA, USA) using an Agilent 2100 Bioanalyzer Chip according to the instructions of the manufacturer. A poly-A pull-down step was performed to enrich the mRNAs from the total RNA samples and proceeded to library preparation by using an Illumina TruSeq RNA prep kit. The libraries were then sequenced using Illumina HiSeq2000 at the Columbia University Genome Center. Multiplex samples with unique barcodes were mixed in each lane, yielding a targeted number of single-end 100 bp reads for each sample as a fraction of 180 million reads for the whole lane. RTA software (Illumina, Inc., San Diego, CA, USA) was used for base calling, and bcl2fastq (version 1.8.4) was used to convert BCL to the fastq format, coupled with adaptor trimming. The reads were mapped to a reference human genome (hg38) using STAR software [43]. 

### 2.7. Differential Expression Analysis 

The RNAseq data were analyzed using iDEP (ver1.0) (http://bioinformatics.sdstate.edu/idepg/, accessed on 12 August 2022) [44,45]. Differential expression analysis was conducted using DESeq2 [46], an R package based on a negative binomial distribution that models the number of reads from RNA-seq experiments and tests for differential expression. Differentially expressed genes (DEGs) between the mutant (DEL) and control (WT) samples and DEGs with or without imatinib treatment with genotype as a covariant were determined statistically with false discovery rate (FDR) correction. A list of significant DEGs was defined by an FDR *p*-value of <0.05 and a fold change of >1.2. The expression changes in the genes within the deletion regions were used as positive controls for experimental validation.

### 2.8. Gene Ontology Enrichment and KEGG Pathways Analysis

To determine if the DEGs shared some common biological processes, molecular function, or cellular components, the enrichment of the Gene Ontology terms was tested using shinyGO [47] in the iDEP with default settings. The gene lists of up-regulated and down-regulated DEGs for both the genotype and imatinib treatment were analyzed separately. To determine if the DEGs were enriched in physiological pathways, GAGE KEGG pathway analysis was conducted with a threshold of FDR *p* < 0.05 [48]. The gene expression pattern data of the DEGs that are involved in a specific KEGG pathway were mapped with Pathview [49,50]. 

### 2.9. Treatment with Imatinib

Imatinib was purchased from Tocris and was dissolved in distilled water as instructed (https://www.tocris.com/products/imatinib-mesylate_5906, accessed on 7 December 2022). Aman et al. [51] applied 10 μM imatinib to rescue the thrombin-induced leaky monolayer formed by human lung microvascular and human umbilical vein ECs. Treatment for 2 h can bring back the TEER of both disrupted EC monolayers. Treatment with 10 μM imatinib for 2 h also reduces the increased permeability to Dex-40k of human microvascular EC monolayers by vascular endothelial growth factor (VEGF) [17]. Therefore, we applied similar doses of imatinib and durations as reported in [17,51] in our study. To investigate the effects of imatinib on TEER, permeability to Dex-70k, tight junction proteins, and the glycocalyx of iBBB, after iBBB was formed, 2 μM, 10 μM or 50 μM imatinib was added into the medium for 1 h or 2 h. For Western blotting and RNAseq, 10 μM imatinib was added to the medium for 2 h. 

### 2.10. Statistical Analysis

For solute permeability and TEER measurements, the data were presented as mean ± standard deviation (SD); for tight junction proteins and glycocalyx (HS intensity) assays, the data were presented as mean ± standard error (SE). The Wilcoxon matched-pairs signed rank test was used for the comparisons between the control and 22q11.2DS data. A *t*-test or two-way ANOVA was used for the comparisons between the treatments and non-treatments. Kurtosis analysis was used to compare the intensity distribution profiles of the junction proteins for the control and 22q11.2DS cases, as well as with and without imatinib treatments. *n* ≥ 6 samples for each case for the solute permeability and TEER experiments. *n* = 3 samples for each case for junction proteins and EG quantification. The samples were from at least 3 independent differentiations. For RT-qPCR, the data were presented as mean ± standard deviation (SD). Student *t*-test was used for the comparison between 22q11.2DS and the control data, as well as with and without imatinib treatments. For Western blotting, three biological samples per condition (3 individual patients and 3 individual controls) were tested with three independent experimental replicates. The bands were quantified with ImageStudio Lite software (Licor Inc., version 5.0). The ratios of phosphorylated CRKL/total CRKL band intensities were used to determine the active status of CRKL. To control for the batch effect between experimental replicates, a linear mixed model was used to determine the statistical significance using the lme4 R package. A level of *p*-value < 0.05 was considered statistical significance in all the experiments. 

## 3. Results

### 3.1. Imatinib Restores the Barrier Function of 22q11.2DS iBBBs

Imatinib was employed to diminish the increased EC permeability in sepsis and systemic capillary leak syndrome [17,51]. Therefore, we tested if imatinib can rescue the barrier function deficits in 22q11.2DS iBBBs. We differentiated three pairs of 22q11.2DS and control iPSC lines into HBMECs and generated iBBBs from these HBMECs [5]. We first evaluated the effect of imatinib on the TEER of iBBBs. The TEER reflects the barrier function to ions and small molecules. The reciprocal of the TEER (1/TEER) represents the permeability of iBBB to ions or small molecules. Figure 1 demonstrates that for three paired iBBBs, the TEER of 22q11.2DS iBBBs is only 64–73% that of the corresponding WT iBBBs, which is consistent with our and other previous studies [5,52,53,54]. Interestingly, treatment with 2, 10, and 50 µM of imatinib for 1 h and 2 h can bring back the decreased TEER in the 22q11.2DS iBBB to the level of the WT iBBB. However, we did not see any effects of imatinib on the TEER of WT iBBBs within the tested dose range and duration. We then evaluated the effect of imatinib on the permeability of iBBBs to a large molecule, Dex-70k. Figure 2A shows that compared to the WT iBBB, the permeability to Dex-70k of the 22q11.2DS iBBB is higher, about 1.3–1.5-fold that of the WT iBBB for the three paired iBBBs. The treatment of imatinib at 2, 10, and 50 µM imatinib for 2 h can normalize the increased permeability of the 22q11.2DS iBBB to a comparable level as seen in the WT iBBB. The same as for TEER, this dose and treatment duration of imatinib does not influence permeability to Dex-70k of the WT iBBB. However, different from the effect on TEER, 1 h treatment with 10 µM imatinib does not reduce the increased permeability to Dex-70k of the 22q11.2DS iBBB to that of the WT iBBB (Figure 2B). The effect of imatinib on permeability to large molecules is time-dependent because the major barrier to small and large molecules in the BBB is different [55]. Our results indicate that imatinib can rescue the deficits in the barrier function of 22q11.2DS iBBBs.

### 3.2. Imatinib Restores the Tight Junctions and Endothelial Glycocalyx of 22q11.2DS iBBBs

The barrier function of the BBB is determined by its structural components. As indicated before [55,56,57], the tight junctions are the major barrier to ions and small molecules, and the endothelial glycocalyx is the major barrier to the large molecules, which is compromised in the iBBBs of 22q11.2DS [5]. Therefore, we investigated the effect of imatinib on the tight junctions and the endothelial glycocalyx of the iBBB. Figure 3 shows the effect of imatinib on the tight junction proteins of ZO-1 and claudin-5 (CLN-5). Without imatinib treatment, the intensity profiles of ZO-1 and CLN-5 at 22q11.2DS iBBBs are lower than those of WT iBBBs in the three paired iBBBs, indicating that those tight junctions are compromised in the deletion iBBB. Consistent with the effect on TEER, treatment with imatinib at 2, 10, and 50 µM imatinib for 2 h and 10 µM for 1 h (data not shown) can rescue the compromised ZO-1 and CLN-5 to the comparable levels of WT iBBB. Figure 4 shows the effect of imatinib on the endothelial glycocalyx (HS) of iBBBs. Without imatinib treatment, the intensity of HS at 22q11.2DS iBBBs is only 25–55% that of WT iBBBs in the three paired iBBBs, indicating the disrupted glycocalyx in the deletion iBBB. Consistent with the effect on permeability to Dex-70k, treatment with imatinib at 2, 10, and 50 µM for 2 h can recover the glycocalyx at the deletion iBBB to the level of the WT iBBB, but 10 µM for 1 h cannot. These results indicate that imatinib can restore the tight junctions and endothelial glycocalyx of 22q11.2DS iBBBs. 

### 3.3. Imatinib Increases Active Forms of CRKL in HBMECs

To test our hypothesis that imatinib leads to a decrease in phosphorylated CRKL (inactive form) through Abl inhibition and increasing the active form of CRKL, we collected HBMECs with and without 10 µM-2 h imatinib treatment and performed Western blot analysis to determine the levels of the total and phosphorylated CRKL proteins. Our results indicate that the total and pCRKL levels are significantly lower in 22q11.2DS HBMCs than in the controls (*p* = 8.8 × 10^8^ for total CRKL and *p* = 0.0012 for pCRKL, linear mixed model ANOVA test) (Figure 5A,B). Our statistical analysis on all three pairs of samples shows that imatinib has no significant effect on total CRKL protein level (*p* > 0.21, linear mixed model ANOVA test), while it has a strong inhibitory effect on phosphorylated CRKL protein levels (*p* < 0.006, linear mixed model ANOVA test) in both 22q11.2DS and wild type HBMECs as expected (Figure 5B). Thus, the ratio of inactive (phosphorylated CRKL) to total CRKL is significantly lower after imatinib treatment (*p* < 1.8 × 10^−4^, linear mixed model ANOVA test), suggesting that there are more active forms of CRKL in 22q11.2DS HBMCs after imatinib treatment (Figure 5C). These findings imply that imatinib can alter CRKL phosphorylation status via its inhibitory effects on Abl kinase.

### 3.4. Transcriptomic Changes of iBBBs with Imatinib Treatment

To unbiasedly determine the molecular mechanisms associated with imatinib-mediated rescue, we conducted bulk RNA sequencing (RNAseq) on three pairs of iBBB BMECs with and without 10 µM-2 h imatinib treatment. We first examined the transcription difference due to genotype. As expected, the genes within the deletion region are among the most significantly down-regulated genes, indicating that the experimental system worked well (Figure 6A left, Appendix A). GO analysis indicated that the up-regulated genes are associated with junctions, while the down-regulated genes are associated with cell–cell communication (Appendix A). These results agree well with our previous analysis [5], indicating the deficits in junction function in the 22q11.2DS iBBB. We then determined the differentially expressed genes (DEGs) in response to imatinib treatment with genotype as a covariate. There are 473 up and 349 down-regulated genes in response to imatinib treatment (Figure 6A right, Appendix A). We found that the down-regulated genes are associated with enriched GO terms, including Mitotic spindle organization, Microtubule cytoskeleton organization involved in mitosis, and Mitotic nuclear division, while the up-regulated genes are enriched for GO terms such as phosphorylation, cellular protein modification processes, and protein modification processes (Figure 6B, Appendix A). The effect of imatinib on the inhibition of cell division is consistent with the fact that it has been used as an anti-cancer drug in clinics. The effect of imatinib on up-regulating cellular protein modification genes is an interesting novel finding. Therefore, we further analyzed the involvement of these DEG genes that are associated with KEGG pathways in response to imatinib treatment using the GAGE analysis. Interestingly, the top significantly (FDR *p* < 0.05) up-regulated pathways are focal adhesion (FDR *p* = 3.8 × 10^−3^); PI_3_K-Akt signaling pathway (FDR *p* = 1.2 × 10^−2^) and tight junction (FDR *p* = 1.4 × 10^−2^) (Table 1). When mapping the altered gene expression levels onto the genes in these affected KEGGs pathways, we found that the CRKL signaling pathway, which is involved in the focal adhesion pathway and down-regulated in the 22q11.2DS iBBB, was up-regulated after imatinib treatment (Figure 6C). Similarly, genes including ZO-1, claudins and occludin, which are down-regulated in the tight junction pathway in the 22q11.2DS iBBB, were up-regulated after imatinib treatment (Figure 6D). These results strongly support that the CRKL signaling and tight junction pathways are the key effectors that imatinib acts on the 22q11.2DS BBB. Combined with the functional and structural results at cellular levels, our study pinpointed a new mechanism that imatinib improves the tight junction protein assembly in the 22q11.2DS BBB via the Abl/CRKL. 

## 4. Discussion

In this study, we showed that imatinib, an inhibitor of Abl kinase, can rescue the deficits in the iBBBs of 22q11.2DS patients. We further demonstrated that imatinib could increase the active form of CRKL, a key effector of Abl kinase. Finally, through RNAseq analysis, we showed that imatinib could enhance the CRKL signaling pathways and improve the gene expression of the key genes in tight junction pathways, such as claudins, occludin, and ZO-1. The underlying mechanism is illustrated in Figure 7 as a working model. 

The connection between imatinib and inflammatory agents and VEGF-mediated endothelial permeability has been explored previously. For example, imatinib was used to protect against endothelial barrier dysfunction induced by thrombin and histamine [17,51] and by VEGF [17] in the monolayers of human lung microvascular ECs, human umbilical vein ECs, and human microvascular ECs. In addition, several studies indicated that imatinib preserves blood–brain barrier integrity and reduces brain injury in murine models [58,59]. However, no study has been conducted before on the effect of imatinib on the human BBB structure and function. Our current study demonstrates that imatinib could be a drug that can potentially rescue the barrier function of the compromised human BBB, at least partially through the proposed molecular mechanism.

Chislock et al. [17] investigated the molecular mechanism by which thrombin, histamine, and VEGF increase endothelial permeability and that by which imatinib decreases permeability. They found that Abl kinases are activated by these permeability-enhancing factors to increase phosphorylated CRKL (pCRKL), while pretreatment with imatinib can abolish the increased level of pCRKL. These results agree well with our observation that the Abl/CRKL signaling pathway is a key molecular mechanism underlying the effects of imatinib. Our results further indicate that imatinib promotes the barrier function by enhancing the production of tight junction proteins such as ZO-1, claudins, and occludin and their assembly into junction sites. In addition, our study provides much more detail on the molecular mechanism for understanding the anti-vascular leak effect of imatinib.

Other features revealed by our RNAseq analysis include confirmation of the inhibitive effects of imatinib on HBMEC cell division and cell cycling by targeting mitotic spindle organization. These findings have multiple implications for clinical practice. On the one hand, imatinib can improve the permeability feature of HBMCs, while it might interfere with angiogenesis. It is thus suggested that when imatinib is applied to prevent vascular leaks in clinical settings, the side effects of inhibiting angiogenesis and repair recovery should also be taken into consideration. Since imatinib (Gleevec) has been applied to treat a variety of cancers for more than twenty years [22], its therapeutic effects should be significantly over side effects for treating chronic diseases. It is known that neurodevelopment has a critical window (in days instead of years). If the rescue of the compromised BBB benefits neurodevelopment through its impact on the critical window, imatinib treatment might be an option for alleviating the symptoms. However, further investigations are needed for this perspective.

One interesting observation relating to the effects of imatinib is that although imatinib increases TEER and decreases permeability to Dex-70k of the 22q11.2DS iBBB, as well as restores its tight junctions and endothelial glycocalyx to the comparable level of the control iBBB, it does not alter the barrier function and structural components of the control (healthy) iBBB. While treatment with imatinib at 2, 10, and 50 µM for 1 h can recover the tight junctions and TEER, treatment for a longer time, 2 h, can rescue the endothelial glycocalyx and permeability to Dex-70k of the compromised iBBB. These findings indicate that the rescue has a dose window of 2–50 µM for imatinib. The recovering time is also different for different structural components of the BBB, e.g., tight junctions and glycocalyx. Our observations are consistent with prior studies that the recovery of the endothelial barrier function is time-dependent with imatinib at 10 µM, as reported by Chislock et al. and Aman et al. [17,51].

In our current study, the TEER of WT iBBBs ranges from 143 to 171 Ω cm^2^, which is much less than that of the WT iBBB generated from hiPSCs in previous studies [32,60]. To find the reason for this discrepancy, we repeated the generation of WT iBBBs by using the same seeding density as that in [32]. Appendix A shows that a much higher seeding density used in [32], 1 million/cm^2^, can achieve a TEER as high as ~630 Ω cm^2^ after culturing for one day, and ~4300 Ω cm^2^ after 2 days, the same as in [32]. However, for the DEL iBBB, the much higher seeding density of 1 million/cm^2^ did not generate the higher TEER for the DEL iBBB after culturing for 1 and 2 days, compared to our seeding density, 50 k or 0.05 million/cm^2^. Furthermore, many cells were dead in the DEL iBBB after culturing for 3 days at 1 million/cm^2^ seeding density. Instead, our low seeding density of 50 k/cm^2^ can generate stable TEER in 6–7 days for DEL iBBBs as well as for WT iBBBs. Therefore, for the purpose of our current investigation and rescue the compromised DEL iBBB, we used a seeding density of 50 k/cm^2^, although the TEER of the WT iBBB is not as high as in [32], with a high seeding density of 1 million/cm^2^. Compared to the WT iBBB formed in 2 days at a seeding density of 1 million/cm^2^, iBBB formed in 6 days at a seeding density of 50 k/cm^2^ has similar EC morphology and arrangements, comparable claudin-5 but reduced ZO-1 expressions (Appendix A). Appendix A show the titration of cell seeding density and the change in the TEER of the iBBB as a function of culturing time for both WT and DEL iBBBs. These experiments showed that the seeding density of 50 k/cm^2^ and culturing time of 6–7 days are optimal in our study.

## 5. Conclusions

Our study indicates that the inhibition of Abl kinase by imatinib can restore the integrity and function of the compromised BBB in 22q11.2DS patients. Imatinib increases active forms of CRKL, as shown by Western blotting and a transcriptome study, and enhances the genes responsible for cell adhesions. It also promotes the production of tight junction proteins, as shown by the quantitative measurement of immunostaining images and pathway analysis.

## Figures and Tables

**Figure 1 cells-12-00422-f001:**
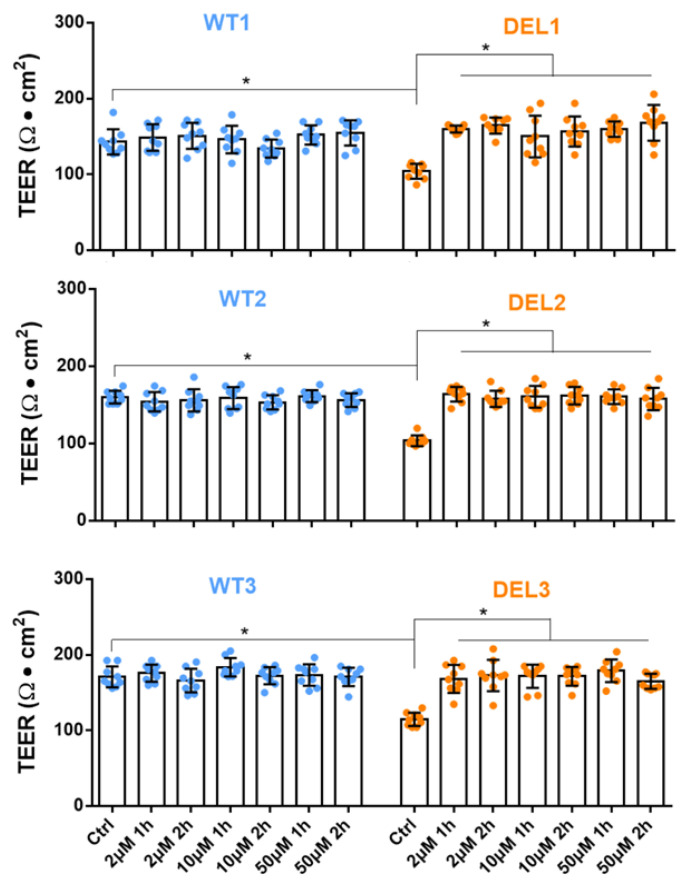
Effects of imatinib on TEER of iBBB for control (WT) and 22q11.2DS (DEL). Three paired control and 22q11.qDS samples were tested. * *p* < 0.05. Values are mean ± SD. *n* ≥ 6 for each case were from at least 3 independent experiments.

**Figure 2 cells-12-00422-f002:**
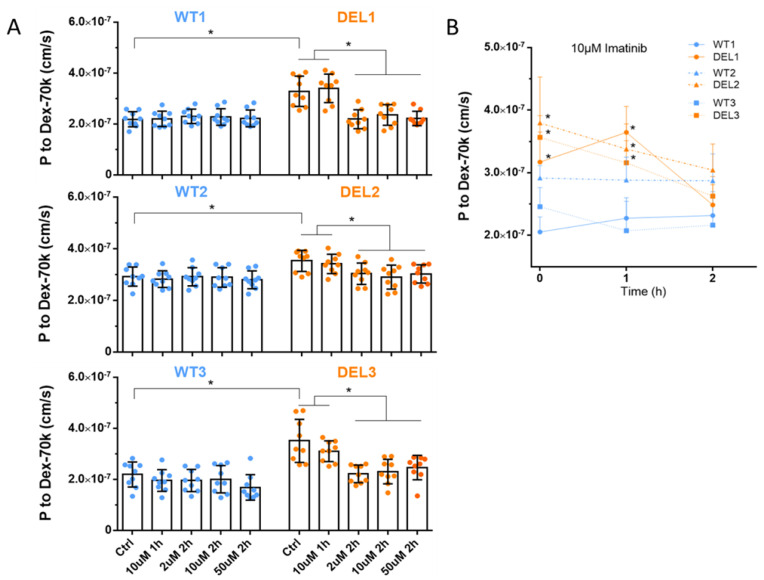
Effects of imatinib on permeability of iBBB to Dextran-70k for control (WT) and 22q11.2DS (DEL). (**A**) Three paired control and 22q11.qDS samples were tested. (**B**) Time-dependent effects of 10 µM imatinib on P to Dex-70k. * *p* < 0.05. Values are mean ± SD. *n* ≥ 6 for each case were from at least 3 independent experiments.

**Figure 3 cells-12-00422-f003:**
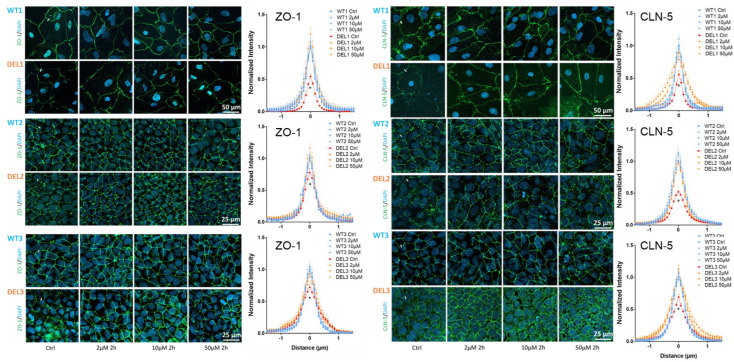
Effects of imatinib on tight junction proteins of iBBB for control (WT) and 22q11.2DS (DEL). Confocal images showing tight junction proteins (ZO-1 and claudin-5) at the iBBB for the control and 22q11.2DS from 3 paired iPSCs. The line plots show the comparison of the intensity profiles of ZO-1 and claudin-5 labeling along a ~3 μm line perpendicular to the EC junctions (white lines in the confocal images) between control (blue lines) and 22q11.2DS (red and orange lines). The peak intensity of ZO-1 or claudin-5 labeling from the control of WT was used for the normalization. *n* = 3 samples with 60–90 junctions (240–360 perpendicular lines) for each case were averaged for each line in the plot. * *p* < 0.05. Values are mean ± SE.

**Figure 4 cells-12-00422-f004:**
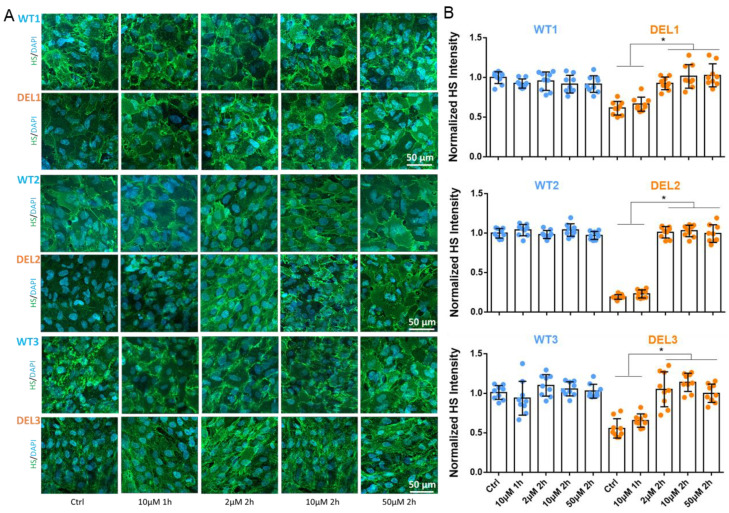
Effects of imatinib on endothelial glycocalyx (EG) at iBBB for control (WT) and 22q11.2DS (DEL). (**A**) Confocal images showing heparan sulfate (HS) of EG at iBBB for the control and 22q11.2DS from 3 paired iPSCs. (**B**) Normalized HS intensity at iBBB for the control and 22q11.2DS. *n* = 3 samples with 9 fields (each field 320 µm × 320 µm) analyzed for each case, * *p* < 0.05. Values are mean ± SE.

**Figure 5 cells-12-00422-f005:**
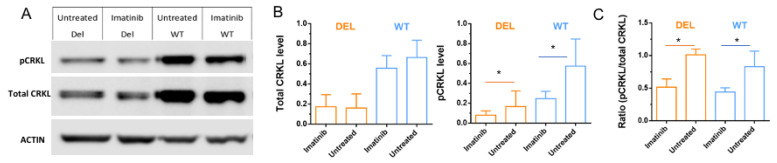
Imatinib increases un-phosphorylated (active form) CRKL in HBMECs. (**A**) Western blot shows decreased expressions of pCRKL in HBMECs after imatinib treatment. (**B**) Statistical analysis of total and phosphorylated CRKL indicates that the protein level of total CRKL is not affected by imatinib treatment (*p* > 0.21, linear mixed model ANOVA test), while pCRKL level is significantly reduced (* *p* < 0.006, linear mixed model ANOVA test). (**C**) Comparison of pCRKL to total CRKL indicates that this ratio is significantly decreased after imatinib treatment (* *p* < 0.001).

**Figure 6 cells-12-00422-f006:**
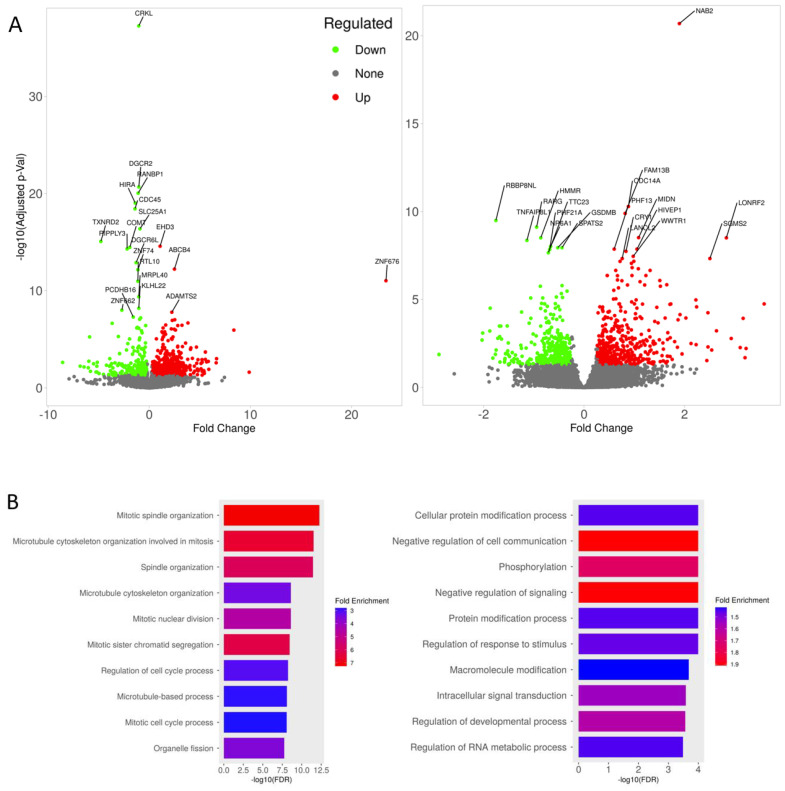
Transcriptomic changes by imatinib treatment. (**A**) The volcano plots of DEGs without (**left**) and with (**right**) imatinib treatment. The green spots indicate the down-regulated genes, and the red spots indicate those up-regulated (FDR *p* < 0.05, fold change > 1.2). (**B**) GO terms that are enriched in down-regulated (**left**) and up-regulated (**right**) of the DEGs with imatinib treatment. (**C**) Expression patterns of the genes in the CRKL signaling pathway involved in the focal adhesion were down-regulated in 22q11.2DS but up-regulated after the imatinib treatment (inset). (**D**) Expression patterns of the tight junction genes that are involved in the tight junction pathway were down-regulated in 22q11.2DS but were up-regulated after the imatinib treatment (inset).

**Figure 7 cells-12-00422-f007:**
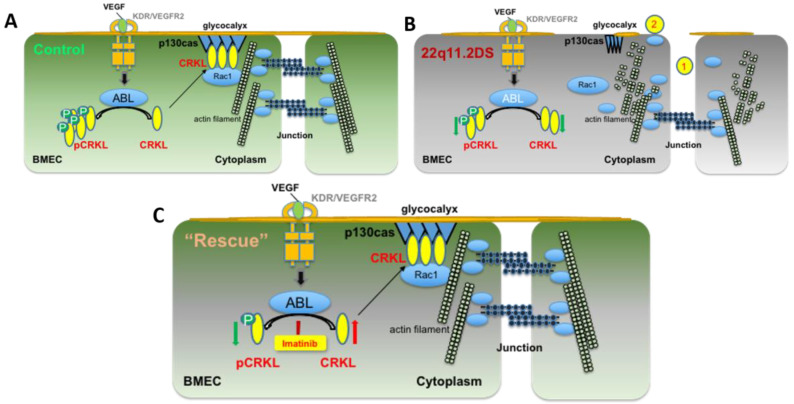
Diagram showing the possible molecular mechanism for the restoration of the BBB integrity by imatinib. Signaling pathways of Abl, CRKL and their relationship with endothelial junction and glycocalyx formation in the normal control BBB (**A**) and in the 22q11.DS BBB (**B**), in which haploinsufficiency of CRKL leads to lower level of unphosphorylated (active form) CRKL and induces interference with tight junctions and glycocalyx formation; (**C**) Inhibition of Abl kinase activity by imatinib increases the level of active form CRKL proteins and their interactions with p130Cas proteins to restore tight junctions and glycocalyx close to the control level.

**Table 1 cells-12-00422-t001:** Up-regulated pathways by imatinib.

Direction	GAGE Analysis: IMATINIB vs. CONTROL	Statistic	Genes	Adj. *p*-Value
Up	Focal adhesion	4.28	168	3.8 × 10^−3^
	PI_3_K-Akt signaling pathway	3.8134	257	1.2 × 10^−2^
	Human cytomegalovirus infection	3.5472	153	2.3 × 10^−2^
	Tight junction	3.4354	128	2.4 × 10^−2^
	Human papillomavirus infection	3.3749	259	2.4 × 10^−2^
	Pathways in cancer	3.3199	408	2.4 × 10^−2^
	Salmonella infection	3.246	208	2.6 × 10^−2^
	Prostate cancer	3.2287	82	2.6 × 10^−2^
	Cytokine-cytokine receptor interaction	3.2113	103	2.6 × 10^−2^
	Kaposi sarcoma-associated herpesvirus infection	3.1396	133	2.6 × 10^−2^
	Amyotrophic lateral sclerosis	3.1259	299	2.6 × 10^−2^
	Regulation of actin cytoskeleton	3.1104	166	2.6 × 10^−2^
	Hepatitis C	3.0765	114	2.8 × 10^−2^
	Alzheimer disease	2.9958	311	3.2 × 10^−2^
	JAK-STAT signaling pathway	2.9416	84	3.6 × 10^−2^
	Leukocyte transendothelial migration	2.9201	83	3.6 × 10^−2^
	Ras signaling pathway	2.8883	166	3.6 × 10^−2^
	Prion disease	2.8792	213	3.6 × 10^−2^
	Small cell lung cancer	2.7779	84	4.7 × 10^−2^
	Proteoglycans in cancer	2.7667	170	4.7 × 10^−2^

## Data Availability

Data are contained within the article or Appendix A.

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
