# Peer review of "Inhibition of Abl Kinase by Imatinib Can Rescue the Compromised Barrier Function of 22q11.2DS Patient-iPSC-Derived Blood–Brain Barriers"

_cells, 2023, doi:10.3390/cells12030422_

Round 1
Reviewer 1 Report
This manuscript takes three iPSC lines from individuals with 22q11.2DS and age matched WT controls and differentiates them into iBBB models. These are then cultured on transwells and used to preform TEER and large molecule permeability experiments. Cells are also treated with imatinib an ABl kinase inhibitor, and the manuscript shows that this is able to partially rescue the cell phenotype. Increasing TEER, tight junction, and glycocalyx while decreasing 70kDa dextran permeability. This repair is accompanied by a hypothesis of the underlying mechanism. The manuscript suggests that this treatment strategy may be effective for BBB breakdown, but fails to show how this strategy is effective outside of this specific mutation.
Major comments
From the methods it is unclear if the Y-27623 was in the culture for the full length or just part of the experiment. If it was included for an extended period during the culture of the cells and the final experiments it raises some questions as to the accuracy and relevancy of the results as the Rho-kinase pathway has important interplay with focal adhesions.
Similarly it would be interesting to note if the DELx cells required the ROCK inhibitor to form monolayers.
The final paragraph stretching from 409-422 is interesting to note for these cells. It would be beneficial to move this paragraph to the methods or at least reference it there, as individuals familiar with this differentiation may question the results presented as this TEER is very low for cells cultured with RA. The authors should show the titration of cell density that resulted in this final seeding density.
The low TEER does imply that the barrier that the WT iBBB is forming may be disrupted to some degree and may have benefited from imatinib, but did not, suggesting that this may not be applicable for all BBB disruption. Line 376/7 suggests that imatinib could rescue barrier function in compromised BBB – is there any evidence that this is a global rescue or more specific for this genetic mutation?
One of the reasons that iBBB models are typically used after 2 days is that some important phenotype characteristics are found to begin to decline, this is consistent with the de-differentiation of BMECs ex-vivo. Can thee authors verify that the phenotype of the d6 WT iBBB is comparable to d2 iBBB models seeded at higher density?
The authors use 70kDa dextran as the tracer molecule, which that the WT iBBB may not have a very strong monolayer to begin with. The authors should consider repeating key experiments using a small molecule tracer such as sodium fluorescein, lucifer yellow, or even a smaller MW dextran. This is alluded to in line 281.
The implication in 394-6 that imatinib has therapeutic effects greater than it’s side effects is seen in cancer treatment and obviously strongly depends on the severity of the disease that is attempting to be treated. In this work you suggest using a chemotherapeutic for patients with this mutation, who granted have significant symptoms, but they are not acute symptoms but rather chronic. The authors mention that this should be considered (393), but do not show any results or make arguments that justify that this specific signaling pathway may be relevant in any other situation.
Minor
Please clarify if three independent experiments in 177 signifies three separate wells from the same differentiation or three separate differentiations. Similarly for RNA isolated for RNAseq.
Figure 3 is difficult to distinguish between the different shapes in the traces of the junctional intensity, particularly the DELx ctrl from the treated conditions.
It would be beneficial if the authors would pull a table of key genes (ZO1, CLDN5, OCCLN) that are referenced in the text to show the relative fold change across different conditions.
Reviewer 2 Report
Li et al. performed an interesting study showing that imatinib can potentially rescue a diminished blood-brain barrier (BBB) by its ability to activate CRKL via phosphorylation and therefore activating the Abl/CRKL signalling pathway. The authors demonstrated changes in the permeability of an induced BBB by usage of hiPSC-derived human brain microvascular endothelial cells (HBMECs), whereby the generated BBB by the iPSC of 22q11.2 DS (Deletion Syndrome or DiGeorge Syndrome) patients is compromised. The authors employed different methods to show this, including confocal microscopy, Western Blotting and RNAseq.
The study is of interest for the readership, however there are some issues which need to be addressed before the study should be published.
1. The title of the manuscript should reflect the results more accurately, i.e. only a partial rescue is possible, please revise “…can partially rescue the compromised …”.
2. There are several sentences in the introduction, the material and methods section and in the results that are copied one-to-one from other already published material. Whereas it is questionable to have it in the material and methods section, it is strongly recommended that the authors revise that in the other sections to avoid the accusation of plagiarism.
For instance, line 42 “Schizophrenia patients require long-term medical care and impose a major economic burden on the healthcare system.”, is copied from Li et al., 2021, published in Cells; or line 51 “a set of small GTPases, such as RAC1 and RAP1, which are critical in regulating cell adhesion and junction stability”, copied from the same source Li et al., 2021, Cells; or line 253 figure legend.
3. In the diseased cells, CRKL expression is reduced leading to a compromised BBB. At the same time, CRKL is phosphorylated and thus inactivated by Abl kinases. However, if Abl stopps phosphorylation of CRKL, it does not mean that the level of CRKL will increase, the remaining CRKL will just be active again. However, this would not entirely explain the restoration of the BBB , as the levels will not increase. How can the authors explain this discrepancy?
4. In the same vein, the authors show in Fig 5 the levels of total CRKL and phosphorylated CRKL in WT and deletion. I cannot see from the raw Western Blot, that the intensity of the bands change and how the authors can conclude from this experiment that the ratios changes significantly. Could the authors please provide the Western blots that lead to the conclusion that the ratio changes significantly? And how many samples were analysed? How many biological replicates? This is also not clear from the figure legend.
55. The authors did not comment on which vehicle was used for the treatment with the kinase inhibitor imatinib. How is it dissolved and have the authors used it as a control? It is not clear from the figures and the figure legends, whether a vehicle control was included or not. It is necessary to have this control included, in order to distinguish between the effects of the inhibitor and the vehicle.
6. The point mentioned in 5.) would be even more important for RNAseq. The authors did not comment on a vehicle control, they just stated “… we conducted bulk RNA sequencing (RNAseq) on three pairs of iBBB BMECs with and without 10 μM-2h imatinib treatment”. Could the authors please add to this section whether they used a vehicle control and if so, what control and at which concentration. Otherwise, it would be impossible to distinguish between DEGs due to the kinase inhibitor and due to the vehicle control.
7. It would be beneficial, if the authors could actually show the increase of ZO-1, claudins and occluding by means of Western Blotting, as this would give a more accurate picture of the actual increase. This would also strengthen the RNAseq data, as the RNA level does not mean that the protein level is necessarily increased as well.
8. It is plausible that any drug i.e. imatinib has a certain window, where its effects are independent from the applied doses. Therefore, the point that there is a window where it is at a certain dose-window time-dependent and dose-independent is a generalization, which applies to any drug, so please rephrase this paragraph (line 403).
9. The authors state in the conclusions that “Western blotting and a transcriptome study further show that imatinib increases active forms of CRKL, upregulates genes responsible for cell adhesions and promotes the production of tight junction proteins.” The last conclusion cannot be drawn; as far as I can see Western blotting has not been performed for tight junction proteins and a transcriptome study can only say something about the transcriptome, i.e. the RNAs, but not the protein. Again, it would be mandatory to perform Western Blots of the tight junction proteins as well as other selected candidates from the RNAseq in order to draw these conclusions.
Round 2
Reviewer 2 Report
The authors Li et al. responded to the concerns and provided a substantially improved manuscript. Therefore, I accept it in the present form.